

# Effect of exogenous melatonin on growth and antioxidant system of pumpkin seedlings under waterlogging stress

Zhenyu Liu[1,2], Li Sun[1,2], Zhenwei Liu[1,2] and Xinzheng Li[1,2]

[1] College of Horticulture and Landscape, Henan Institute of Science and Technology, Xinxiang, Henan, Xinxiang, China
[2] Henan Province Engineering Research Center of Horticultural Plant Resource Utilization and Germplasm Enhancement, Xinxiang, Henan, Xinxiang, China

## ABSTRACT

Melatonin regulates defense responses in plants under environmental stress. This study aimed to explore the impact of exogenous melatonin on the phenotype and physiology of 'BM1' pumpkin seedlings subjected to waterlogging stress. Waterlogging stress was induced following foliar spraying of melatonin at various concentrations (CK, 0, 10, 100, 200, and 300 $\mu mol \cdot L^{-1}$). The growth parameters, malondialdehyde (MDA) content, antioxidant enzyme activity, osmoregulatory substance levels, and other physiological indicators were assessed to elucidate the physiological mechanisms underlying the role of exogenous melatonin in mitigating waterlogging stress in pumpkin seedlings. The results indicate that pumpkin seedlings exhibit waterlogging symptoms, such as leaf wilting, water loss, edge chlorosis, and fading, under waterlogging stress conditions. Various growth indicators of the seedlings, including plant height, stem diameter, root length, fresh and dry weight, and leaf chlorophyll content, were significantly reduced. Moreover, the MDA content in leaves and roots increased significantly, along with elevated activities of superoxide dismutase, catalase, peroxidase, and soluble protein contents. When different concentrations of melatonin were sprayed on the leaves post waterlogging stress treatment, pumpkin seedlings showed varying degrees of recovery, with the 100 $\mu mol \cdot L^{-1}$ treatment displaying the best growth status and plant morphological phenotypes. There were no significant differences compared to the control group. Seedling growth indicators, chlorophyll content, root activity, antioxidant enzyme activities, soluble protein content, and osmotic adjustment substance content all increased to varying degrees with increasing melatonin concentration, peaking at 100 $\mu mol \cdot L^{-1}$. Melatonin also reduced membrane damage caused by oxidative stress and alleviated osmotic imbalance. Exogenous melatonin enhanced the activities of antioxidant enzymes and systems involved in scavenging reactive oxygen species, with 100 $\mu mol \cdot L^{-1}$ as the optimal concentration. These findings underscore the crucial role of exogenous melatonin in alleviating waterlogging stress in pumpkins. The findings of this study offer a theoretical framework and technical assistance for cultivating waterlogging-resistant pumpkins in practical settings. Additionally, it establishes a theoretical groundwork for the molecular breeding of pumpkins with increased tolerance to waterlogging.

Corresponding author
Zhenwei Liu, liuzhw@hist.edu.cn

## INTRODUCTION

Waterlogging stress is a type of abiotic stress that significantly decreases oxygen levels in the soil, leading to adverse effects on plant growth, development, and physiological characteristics (*Zhang et al., 2019*). Local hypoxia in the root system hinders crop root growth, reduces root vigor, disrupts the root-crown ratio, and induces a rapid decline in root dry mass (*Zhang et al., 2023*). Additionally, waterlogging stress damages the integrity of cell membranes, leading to increased intracellular malondialdehyde levels. This stress also disrupts the antioxidant systems that minimize the levels of reactive oxygen species (ROS), leading to excessive ROS accumulation (*Huang et al., 2017*). Elevated ROS levels damage plant cells and disrupt crucial physiological processes, ultimately leading to apoptosis. To counteract these effects, plants have a stress defense system that regulates the production and elimination of ROS. Antioxidant enzymes such as superoxide dismutase (SOD), peroxidase (POD), and catalase (CAT) play a vital role in scavenging ROS (*Miller et al., 2009*; *Wang et al., 2016*). Therefore, an effective management strategy is needed to improve the plant antioxidant defense system, increase waterlogging tolerance, and decrease ROS production. Using plant growth regulators, such as auxins, gibberellins, cytokinins, abscisic acid, and melatonin, is the most successful and effective method.

Melatonin is a pleiotropic factor with multiple biological functions in plants, participating in physiological processes such as photosynthesis, seed germination, fruit expansion, root development, and osmoregulation (*Zhao et al., 2022*). Previous findings demonstrated that melatonin plays an important role in regulating plant growth and development and enhancing resistance to abiotic stresses such as drought, high temperature, salinity, heavy metals, and bacterial and fungal diseases (*Zhang et al., 2021*). *Chen et al. (2019)* observed that soaking rice seeds in 100 µmol/L of melatonin significantly alleviated the toxic effect of waterlogging stress. *Ahmad et al. (2022)* concluded that the application of 100 µM melatonin with 0.50 g $KNO_3$ as a seed soaking and foliar application can significantly improve the growth of maize by reducing the detrimental effects of waterlogging stress. *Zhang et al. (2020)* reported that foliar spraying with different concentrations of exogenous melatonin alleviated the damage on soft date kiwifruit caused by low temperatures at 4 °C. *Bai et al. (2023)* found that spraying melatonin at a concentration of 100 µmol·$L^{-1}$ can effectively reduce the damage of low-temperature stress to pumpkin seedlings. *Guo (2020)* discovered that melatonin plays a role in regulating JA accumulation and promoting the production of $H_2O_2$. Additionally, $H_2O_2$ has the ability to regulate jasmonic acid content, thereby playing a role in inducing low temperature resistance in pumpkin rootstocks. There are relatively few studies on the effect of exogenous melatonin on waterlogging tolerance in pumpkin.

Pumpkin (*Cucurbita moschata* D.) is an annual herbaceous plant in the genus Cucurbita (*Han et al., 2020*). Its fruit is visually appealing, characterized by a sweet yet non-greasy taste. Pumpkins are rich in nutrients such as vitamin C, β-carotene, proteins, and carbohydrates. In addition, pumpkins have anti-cancer, hypoglycemic, and hypolipidemic properties (*Wang, Li & Zhang, 2010*). Pumpkin plants exhibit strong adaptability and resilience, thriving in diverse environments. It becomes imperative to explore the mechanisms through which pumpkins overcome the effects of waterlogging stress owing to an increase in occurrences of waterlogging-related agricultural and economic losses. Therefore, in this study, we selected BM1, a flood-tolerant pumpkin variety, as the experimental material to explore the regulatory capacity of melatonin in alleviating the effect of waterlogging stress in pumpkins. The findings of this study will provide a theoretical basis for understanding waterlogging tolerance and the mechanism underlying the role of melatonin in enhancing this tolerance in pumpkins.

# MATERIALS AND METHODS

## Experimental materials

The experimental material in this study was BM1 seedlings (known for strong waterlogging tolerance) (*Qiao, 2022*). The seeds were obtained from the Henan Institute of Science and Technology, Henan, China (N:35°16′54.12″, E:113°56′10.54″). Melatonin was purchased from Beijing Suolaibao Bio-technology Co. Ltd., Beijing, China. The ZhuangZhuang seedling substrate (Peat: vermiculite: perlite is 3:1:1, PH is neutral, $N + P_2O_5 + K_2 \geq 4\%$) obtained from Hebei Peiji Biotechnology Co. Ltd., Beijing, China was used to grow the seedlings. The study was conducted in August 2023 in the seedling room of the College of Horticulture and Landscape Architecture, Henan Institute of Science and Technology.

## Experimental design

The experiment comprised five distinct melatonin concentrations (0, 10, 100, 200, and 300 $\mu$mol·L$^{-1}$) and two treatments, no waterlogging and waterlogging. The treatments were as follows: (1) CK, not waterlogging; (2) T0, waterlogging treatment + 0 $\mu$mol·L$^{-1}$ melatonin; (3) T10, waterlogging treatment + 10 $\mu$mol·L$^{-1}$ melatonin; (4) T100, waterlogging treatment +100 $\mu$mol·L$^{-1}$ melatonin; (5) T200, waterlogging treatment + 200 $\mu$mol·L$^{-1}$ melatonin; (6) T300, waterlogging treatment + 300 $\mu$mol·L$^{-1}$ melatonin. Melatonin leaf spray treatment was administered daily to seedlings with one leaf and one heart, ensuring the water droplets condensed on the leaf surface without dripping. The spraying was conducted once every other day, a total of three times. The waterlogging treatment was implemented using the the double-pot method 12 h after the third melatonin treatment, while maintaining other growth conditions (*Liu, 2020*). After 7 days of waterlogging treatment, growth indices (plant height, stem thickness, fresh weight, dry weight) and chlorophyll content of pumpkin seedlings were measured. Leaves and roots were collected to assess relevant physiological indices, with six plants sampled from each treatment. All experiments were repeated three times.
## Test methods

### Growth indicators

The plant height of pumpkin seedlings was assessed by measuring the distance from the base of the cotyledonary node to the top heart leaf following a methodology described by *Bai et al. (2023)*. The stem diameter of the seedlings was determined using Vernier calipers by measuring the diameter of the cotyledonary node in the direction of the cotyledonary leaf unfolding. To determine the fresh weight, the plants were washed with tap water, rinsed three times with distilled water, dried with absorbent paper, and weighed using an electronic balance. For dry weight determination, the pumpkin seedlings were placed in an oven at 105 °C for 15 min, dried at 75 °C until a constant weight was attained, and weighed using an electronic balance.

### Physiological indicators

(1) Root activity: The root activity was determined by the Solaibao Co., Ltd., Beijing, China. kit. The naphthylamine method (*Gao & Yang, 2020*) was used to weigh 0.2 g root samples, add 1 mL extract, homogenize in ice bath, and centrifuge at 8,000 $g$ 4 °C for 10 min. After 37° dark reaction, the supernatant was taken at 10 min and 3 h, and the reagent was fully mixed. After 20 min of color development at room temperature, the absorbance was measured at a wavelength of 520 nm. (2) Chlorophyll content was determined using the ethanol extraction method (*Li, 2000*). Fresh leaves were cut into small pieces, weighed 0.1 g, and placed in a 25 ml volumetric flask filled with 96% ethanol. The leaves were then extracted at room temperature in the dark for 24 h, with periodic shaking. Absorbance values were measured at 470, 649, and 665 nm until the tissue turned completely white. (3) Malondialdehyde (MDA) content was determined by thiobarbituric acid method (*Gao, 2006*). Weigh 0.5 g grinded leaf samples and add 5 mL 5% TCA, centrifuge at 4,000 r/min for 10 min, then take 2 mL supernatant and add 2 mL 0.67% TBA reaction solution. The solution was boiled for 30 min and centrifuged again after cooling. The absorbance values of the test solution at 450, 532 and 600 nm were measured respectively. (4) Superoxide dismutase (SOD): measured using nitroblue tetrazolium (NBT) reduction method (*Gao, 2006*). Determined by the kit of Solaibao Co., Ltd., Beijing, China about 0.1 g of leaf samples were weighed, added with 1 mL of extract, homogenized in ice bath, and centrifuged at 8,000 $g$ at 4 °C for 10 min. The supernatant was added to the reagent in turn, and the wavelength was adjusted to 470 nm. (5) Peroxidase (POD): Determined by the guaiacol colorimetric method (*Gao, 2006*), determined by the kit of Solaibao Co., Ltd., Beijing, China about 0.1 g of leaf samples were weighed, 1 mL of extract was added, homogenized in ice bath, centrifuged at 8,000 $g$ for 10 min at 4 °C. The supernatant was added to the reagent in turn, and the wavelength was adjusted to 540 nm. (6) Catalase (CAT): Determined by the ultraviolet absorption method (*Li, 2000*), determined by the kit of Solaibao Co., Ltd., Beijing, China about 0.1 g of leaf samples were weighed, 1 mL of extract was added, homogenized in ice bath, centrifuged at 8,000 $g$ 4 °C for 10 min; the supernatant was added to the reagent in turn, and the wavelength was adjusted to 240 nm. (7) Soluble protein content: measured by Coomassie brilliant blue method (*Li, 2000*). Weighed 0.2 g grinding uniform leaf samples, added 10 mL distilled
water, took 2 mL homogenate in a centrifuge tube, centrifuged at 4,000 r/min for 20 min, the supernatant was protein extract. 0.1 mL of protein extract was extracted, 0.9 mL of distilled water and 5 mL of Coomassie brilliant blue G-250 reagent were added, and the mixture was fully mixed. After 2 min, the color was compared at 595 nm and the absorbance value was recorded.

## Data processing

The experimental data were sorted and calculated using Excel 2019 (Microsoft Corp., Redmond, WA, USA), and were subjected to an analysis of variance (ANOVA) using SPSS 20 Statistics (SPSS Inc., Chicago, IL, USA). Comparisons between means were carried out using least significance difference (LSD) test at a $p \leq 0.05$. GraphPad Prism 8.00 was used to illustrate the figures.

## RESULTS

### Changes in phenotypic growth of pumpkin seedlings

Waterlogging stress significantly impeded the growth of pumpkin seedlings (Figs. 1–3). Plant height, stem thickness, fresh weight, dry weight, and root length of pumpkin seedlings decreased by 34.57%, 11.3%, 58.61%, 48.04%, and 22.75%, respectively, compared to CK. External application of different melatonin concentrations exhibited varying effects on the growth of pumpkin seedlings under waterlogging stress. Increasing melatonin concentration initially enhanced and subsequently reduced the plant height, stem thickness, fresh weight, dry weight, and root length of pumpkin seedlings. The optimal growth indexes were recorded at a melatonin concentration of 100 $\mu mol \cdot L^{-1}$, with a plant height of 13.67 cm, stem thickness of 6.62 mm, fresh weight of 22.68 g, dry weight of 1.34 g, and root length of 15.83 cm (Fig. 4). These values represented 45.39%, 23.98%, 126.47%, 76.03%, and 12.83% increase compared to a melatonin concentration of 0 $\mu mol \cdot L^{-1}$.

### Changes in root vitality of pumpkin seedlings

The results revealed that melatonin significantly reduced the root activity of pumpkin seedlings subjected to waterlogging stress. Interestingly, an initial rise followed by a decrease in the overall impact on root vitality was observed as the melatonin concentration increased. The root activity of the seedlings in CK was 1,305.84 $\mu mg/(g \cdot h)$. Notably, at a melatonin concentration of 100 $\mu mol \cdot L^{-1}$, the root activity of pumpkin seedlings under waterlogging stress peaked at 1,087.84 $\mu mg/(g \cdot h)$. The lowest root activity was 588.99 $\mu mg/(g \cdot h)$ observed when a 0 $\mu mol \cdot L^{-1}$ melatonin was used. The root vitality of pumpkin seedlings treated with melatonin at all concentrations exceeded that without melatonin treatment, indicating a beneficial effect of melatonin in mitigating the negative impact of waterlogging stress on root vitality (Fig. 5).

### Changes in chlorophyll contents in pumpkin seedlings

Pumpkin seedlings subjected to waterlogging stress exhibited significant reductions in chlorophyll a, chlorophyll b, total chlorophyll, and carotenoids by 74.8%, 52.8%, 42.2%,

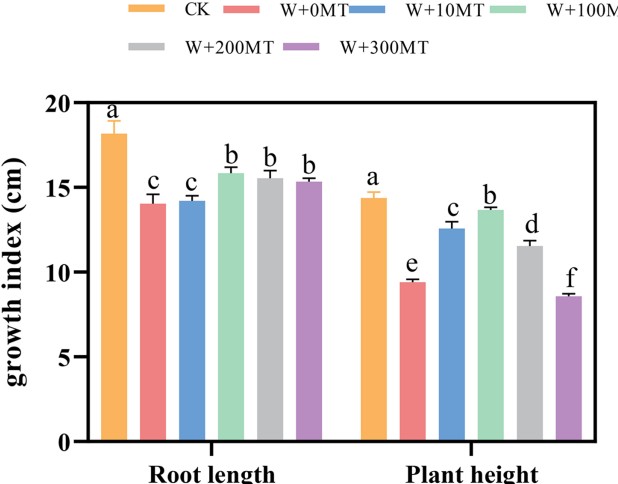

**Figure 1 Growth index (root length, plant height) of pumpkin seedlings under waterlogging stress.** Control 1: not waterlogging (CK), Control 2: waterlogging + 0 μM melatonin (W + 0 MT), Control 3: waterlogging + 10 μM melatonin (W + 10 MT), Control 4: waterlogging + 100 μM melatonin (W + 100 MT) Control 5: waterlogging + 200 μM melatonin (W + 200 MT), Control 6: waterlogging + 300 μM melatonin (W + 300 MT), Data are presented as mean ± SD of three measurements ($n = 3$, biological replicates). Different lowercase letters (a–f) in each column indicate significant differences at $p \leq 0.05$ (least significant difference (LSD) test).

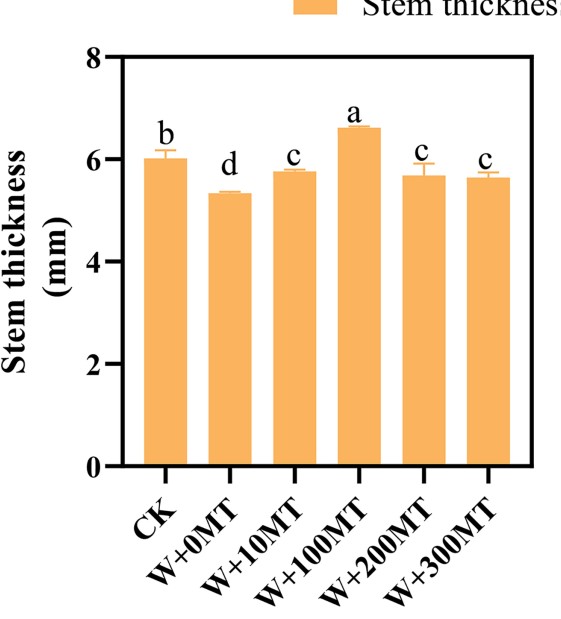

**Figure 2 Growth index (stem thickness) of pumpkin seedlings under waterlogging stress.** Data are presented as mean ± SD ($n = 3$, biological replicates). Different lowercase letters (a–d) in each column indicate significant differences at $p \leq 0.05$ (least significant difference (LSD) test). The abbreviations of treatment names are the same as those described in Fig. 1.

and 77.4%, respectively, compared to CK. The application of varying concentrations of melatonin had a significant impact on the chlorophyll levels in pumpkin seedling leaves. The chlorophyll content in the leaves initially increased and then decreased with an

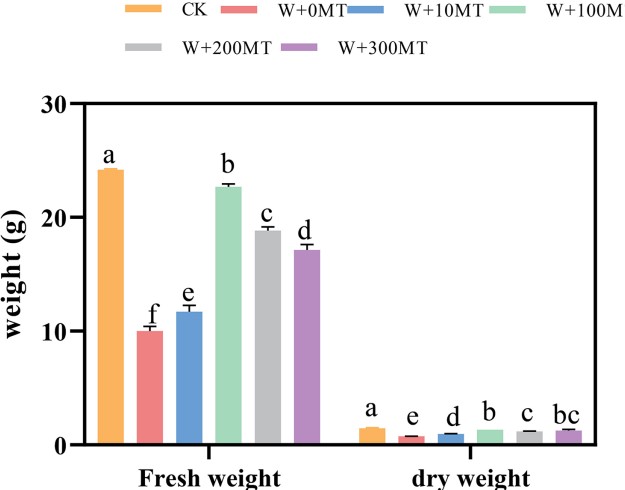

**Figure 3 Growth index (fresh and dry weight) of pumpkin seedlings under waterlogging stress.** Data are presented as mean ± SD ($n = 3$, biological replicates). Different lowercase letters (a–f) in each column indicate significant differences at $p \leq 0.05$ (least significant difference (LSD) test).The abbreviations of treatment names are the same as those described in Fig. 1.

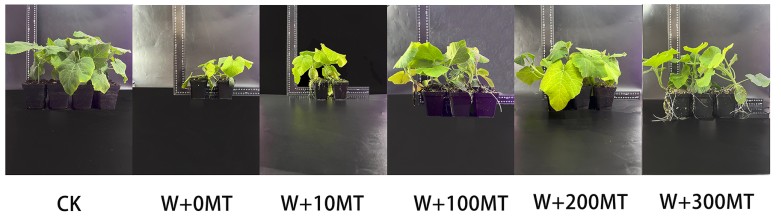

CK     W+0MT     W+10MT     W+100MT     W+200MT     W+300MT

**Figure 4 Plant morphology of pumpkin seedlings under waterlogging stress.** The abbreviations of treatment names are the same as those described in Fig. 1.

increase in melatonin concentration. At a melatonin concentration of 100 µmol·L$^{-1}$, the maximum levels of chlorophyll a, chlorophyll b, and total chlorophyll in pumpkin seedling leaves were 11.56, 5.06, and 16.62 µmol·L$^{-1}$, respectively. At a melatonin concentration of 200 µmol·L$^{-1}$, the highest carotenoid content was 2.18 µmol·L$^{-1}$. chlorophyll a, Chlorophyll b, total chlorophyll, and carotenoid levels in pumpkin seedling leaves increased by 22.7%, 56.7%, 10.4%, and 14.7%, respectively, compared to the control group. This finding indicates that melatonin mitigated the effects of waterlogging stress and increased the chlorophyll content in pumpkin seedlings (Fig. 6).

## Changes in malondialdehyde content in pumpkin seedlings

Waterlogging stress induced a significant increase in malondialdehyde content in both the leaves and roots of the seedlings (Fig. 7). However, the application of melatonin significantly reduced the MDA content in these plant parts. Varying melatonin concentrations showed distinct effects, with the most pronounced reduction in MDA observed in leaves and roots at a concentration of 100 µmol·L$^{-1}$, compared with 0 µmol·L$^{-1}$ corresponding to reductions of 24.57% and 28.82%, respectively. These findings imply that

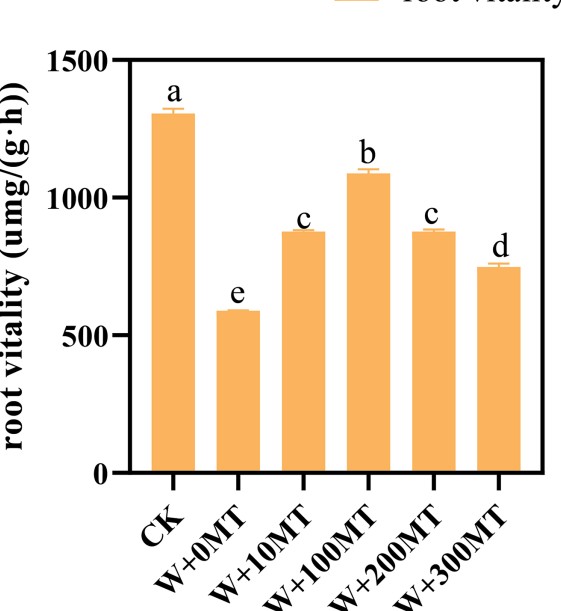

**Figure 5 Root vitality of pumpkin seedlings under waterlogging stress.** Data are presented as mean ± SD ($n$ = 3, biological replicates). Different lowercase letters (a–e) in each column indicate significant differences at $p \le 0.05$ (least significant difference (LSD) test). The abbreviations of treatment names are the same as those described in Fig. 1.

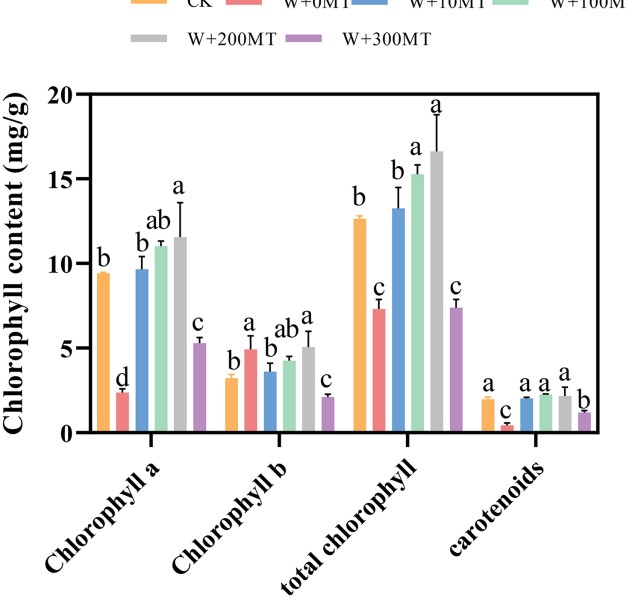

**Figure 6 Chlorophyll contents of pumpkin seedlings under waterlogging stress.** Data are presented as mean ± SD ($n$ = 3, biological replicates). Different lowercase letters (a–d) in each column indicate significant differences at $p \le 0.05$ (least significant difference (LSD) test). The abbreviations of treatment names are the same as those described in Fig. 1.

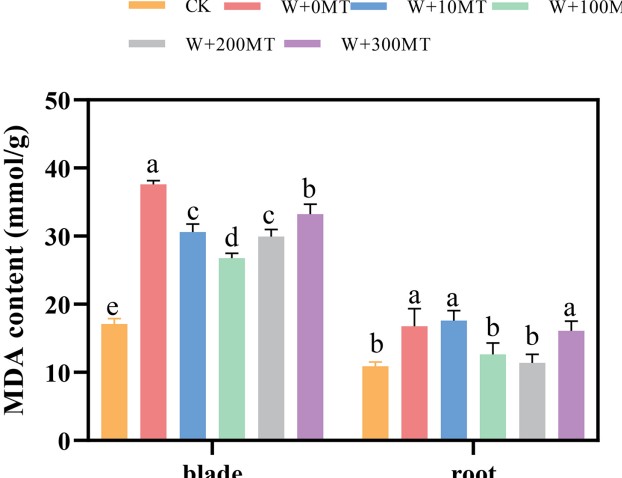

**Figure 7 Malondialdehyde content of pumpkin seedlings under waterlogging stress.** Data are presented as mean ± SD ($n$ = 3, biological replicates). Different lowercase letters (a–e) in each column indicate significant differences at $p \leq 0.05$ (least significant difference (LSD) test). The abbreviations of treatment names are the same as those described in Fig. 1.

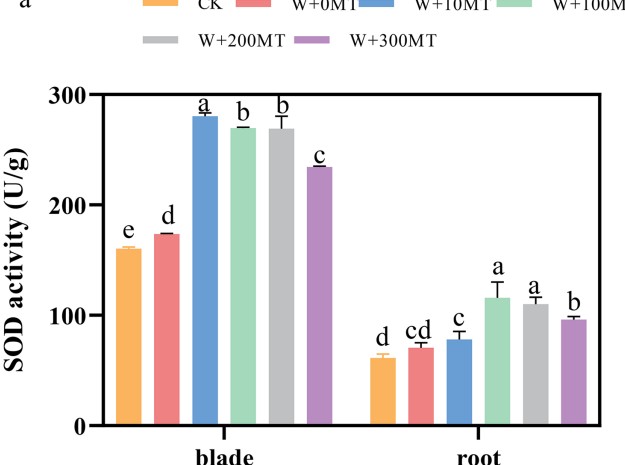

**Figure 8 Activity of antioxidant enzymes (SOD) of pumpkin seedlings under waterlogging stress.** Data are presented as mean ± SD ($n$ = 3, biological replicates). Different lowercase letters (a–e) in each column indicate significant differences at $p \leq 0.05$ (least significant difference (LSD) test). The abbreviations of treatment names are the same as those described in Fig. 1.

melatonin can effectively alleviate membrane lipid peroxidation in pumpkin seedlings under waterlogging stress.

## Changes in the activity of antioxidant enzymes in pumpkin seedlings

The activities of SOD, POD, and CAT enzymes in the leaves and roots of pumpkin seedlings were significantly higher under waterlogging stress than the control group (Figs. 8–10). Following foliar spraying with various concentrations of melatonin, the enzyme activities in both leaves and roots were significantly elevated relative to treatment

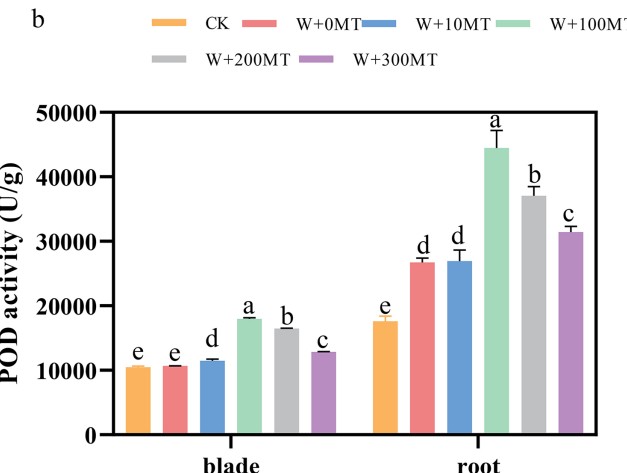

**Figure 9 Activity of antioxidant enzymes (POD) of pumpkin seedlings under waterlogging stress.** Data are presented as mean ± SD ($n$ = 3, biological replicates). Different lowercase letters (a–e) in each column indicate significant differences at $p \leq 0.05$ (least significant difference (LSD) test). The abbreviations of treatment names are the same as those described in Fig. 1.

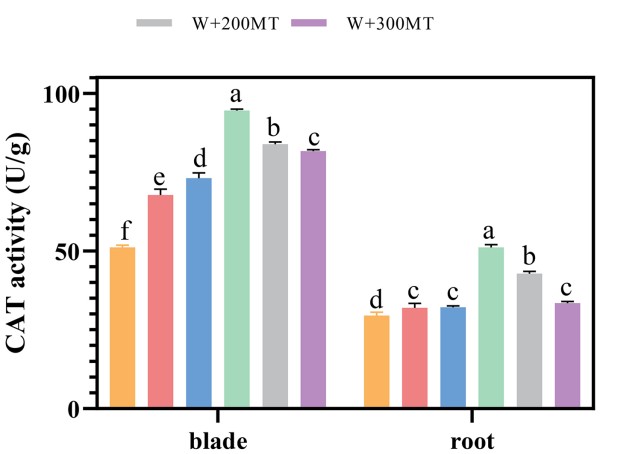

**Figure 10 Activity of antioxidant enzymes (CAT) of pumpkin seedlings under waterlogging stress.** Data are presented as mean ± SD ($n$ = 3, biological replicates). Different lowercase letters (a–f) in each column indicate significant differences at $p \leq 0.05$ (least significant difference (LSD) test). The abbreviations of treatment names are the same as those described in Fig. 1.

with 0 μmol·L$^{-1}$ of melatonin. The enzyme activities exhibited an increasing trend with melatonin concentration, peaking at 100 μmol·L$^{-1}$ for SOD, POD, and CAT in roots as well as for POD and CAT in leaves. The activity of SOD in leaves was highest after application of 10 μmol·L$^{-1}$ melatonin. Notably, the antioxidant enzyme activities in the leaves and roots of pumpkin seedlings were substantially enhanced by 61.41%, 68.46%, and 39.5%, and 64.03%, 66.36%, and 59.81%, respectively, compared to the treatment with 0 μmol·L$^{-1}$ melatonin. These findings indicate that an optimal melatonin concentration can effectively increase the antioxidant enzyme activities in pumpkin seedlings under

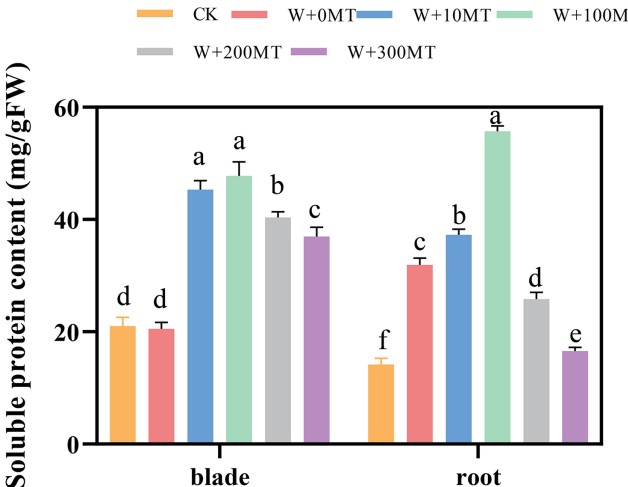

**Figure 11 Soluble protein content of pumpkin seedlings under waterlogging stress.** Data are presented as mean ± SD ($n = 3$, biological replicates). Different lowercase letters (a–f) in each column indicate significant differences at $p \leq 0.05$ (least significant difference (LSD) test). The abbreviations of treatment names are the same as those described in Fig. 1.

waterlogging stress, thereby improving their resilience to such conditions. The most significant impact on the activities of antioxidant enzymes was observed at a melatonin concentration of 100 μmol·L$^{-1}$.

## Changes in the content of the soluble proteins in pumpkin seedlings

The contents of soluble proteins in pumpkin seedling leaves and root systems were significantly higher under waterlogging stress compared to CK (Fig. 11). After foliar spraying with varying concentrations of exogenous melatonin, the contents of soluble proteins in the seedling leaves and root systems were significantly higher than those treated with 0 μmol·L$^{-1}$ melatonin. These levels initially increased and then decreased with increasing melatonin concentration, peaking at 100 μmol·L$^{-1}$. The level of soluble protein in the group treated with 100 μmol·L$^{-1}$ melatonin was 132.57% and 74.39% higher than the 0 μmol·L$^{-1}$ melatonin group. In summary, waterlogging stress increases the content of osmotic molecules in the leaves and roots of pumpkin seedlings. Moreover, the application of an optimal melatonin concentration through foliar spraying substantially elevated the levels of osmotic substances in pumpkins under waterlogging stress, thereby enhancing their resilience to stress.

## DISCUSSION

The response of plants to waterlogging stress is a complex process affecting all stages of plant growth and involving various physiological activities. Waterlogging stress can result in physiological water deficit, the production of reactive oxygen species, disruption of normal plant metabolic activities, damage to cell membrane integrity, dysregulation of the osmotic regulatory mechanism, and ultimately affect plant growth and development (*Li et al., 2022*). Plant growth and development status is a key morphological indicator of their exposure to waterlogging stress. In the present study, waterlogging stress significantly

decreased plant height, stem thickness, root length, and dry fresh weight of pumpkin seedlings and caused leaf wilting and a significant reduction in chlorophyll content. These findings are consistent with previous findings on chrysanthemum reported by *Tao et al. (2024)*. Melatonin, a compound abundantly present in plants, plays a crucial role across various growth and development stages, including enhancing seed germination and delaying leaf senescence, indicating its multifaceted functions in plants (*Bawa et al., 2020*). The application of melatonin through spraying on horticultural crops subjected to biotic and abiotic stresses improves the resistance against these stresses (*He et al., 2022*). *Xu et al. (2024)* discovered that Sunshine Rose Fruit after treating them with 150 µmol/L of exogenous melatonin resulted in increased fruit yields, improved quality, enhanced root zone microecological environment, and effectively slowed down soil quality deterioration. *Muhammad et al. (2023)* have shown that melatonin can effectively alleviate the damage caused by drought stress on maize. Similarly, spraying a 100 mg/L melatonin solution on young grape berries can stimulate fruit growth and expansion (*Meng et al., 2015*). In this study, exogenous melatonin effectively restored normal growth levels in pumpkin seedlings subjected to waterlogging stress. In addition, this treatment significantly increased accumulation of dry mass, potentially by enhancing photosynthesis by increasing the levels of chlorophyll pigments, enhancing reactive oxygen species scavenging capacity, reducing membrane lipid peroxidation, increasing antioxidant enzyme activity, and increasing the content of organic osmotic regulators. The optimal melatonin concentration for foliar spraying to promote pumpkin growth under waterlogging stress was 100 µmol·L$^{-1}$.

The growth and vigor of a plant's root system directly influence the growth of above-ground parts and yield. Waterlogging stress can significantly reduce root vigor, as observed in the pumpkin seedlings in this study. However, foliar spraying with various melatonin concentrations enhanced root vigor in pumpkin seedlings, with 100 µmol·L$^{-1}$ melatonin showing the most significant effect. Research conducted by *Yuan et al. (2022)*. demonstrated that kiwifruit seedlings exhibited reduced root vigor under waterlogging stress, but the application of exogenous melatonin alleviated the damage. Similarly, *Gu et al. (2022)*. observed reduced root vigor in peach seedlings under waterlogging stress, but application of exogenous melatonin alleviated this reduction and partially mitigated the damage to the root system.

*Zhou et al. (2024)* observed that waterlogging stress significantly impaired the photosynthetic efficiency and chlorophyll levels of kale-type oilseed rape leaves. This effect could be attributed to the disruptions in ionic balance, oxidative stress, and metabolic disorders induced by waterlogging stress. Similarly, *Zhou et al. (2023)* found that spraying 100 µmol·L$^{-1}$ melatonin can alleviate the damage of Chinese cabbage caused by high temperature stress. In the current study, pumpkin seedlings exhibited a decrease in total chlorophyll content and the levels of chlorophyll a, chlorophyll b, and carotenoids under waterlogging stress. However, the application of exogenous melatonin alleviated this effect and increased the levels of the chlorophyll pigments. This increase could be attributed to the ability of melatonin to alleviate waterlogging stress, thereby enhancing photosynthesis, promoting the accumulation of dry matter, and increasing plant growth.

Osmoregulation is a vital physiological function in plants that aids them in coping with external stress and maintaining normal growth (*Hua & Li, 2017*). Plants counteract adverse conditions by accumulating osmoregulatory substances. *Yang et al. (2023)* demonstrated that exogenous melatonin increases chlorophyll, soluble sugar, and soluble protein levels in leaves of auberge seedlings under waterlogging stress. In this study, foliar spraying of 100 $\mu$mol·L$^{-1}$ melatonin significantly increased soluble protein content in pumpkin seedlings, enhancing their resistance to waterlogging stress. This effect can be attributed to the stimulation of new protein synthesis in pumpkin seedling leaves by melatonin, enhancing osmoregulation and alleviating cell damage.

Malondialdehyde levels are negatively correlated with the integrity of cell membrane structure. Elevated MDA levels in plants indicate severe membrane damage due to salt stress. Pumpkin seedlings subjected to waterlogging stress produce high levels of $H_2O_2$, leading to oxidative damage, increased membrane permeability, and elevated MDA levels due to membrane lipid peroxidation. In this study, a significant increase in malondialdehyde content was observed in the leaves and roots of pumpkin seedlings under waterlogging stress, consistent with previous findings by *He et al. (2022)*. Melatonin maintains the integrity of the cell membrane, ensuring cell structure stability and enhancing plant tolerance to stress (*Xie, 2022*). Research has demonstrated that foliar application of 100 $\mu$mol/L melatonin can alleviate membrane lipid peroxidation in chrysanthemum seedlings under waterlogging stress, thereby reducing the damage caused by waterlogging (*Tao et al., 2024*). This study revealed a significant decrease in MDA levels in pumpkin seedlings treated with 100 $\mu$mol·L$^{-1}$ melatonin foliar spray.

Waterlogging stress primarily damages plants by disrupting the integrity of plant cell membranes, leading to the accumulation of reactive oxygen species (ROS). To counteract this stress, plants typically rely on antioxidant enzyme systems, such as superoxide dismutase (SOD) and catalase (CAT), to eliminate excess ROS and protect cells from damage. Melatonin, known for its ability to scavenge free radicals, also functions as an antioxidant by enhancing the activity of various enzymes involved in antioxidant defense (*Li et al., 2023*). In the present study, pumpkin seedlings exposed to waterlogging stress exhibited increased SOD and CAT activities, with a further increase observed after application of melatonin.

Exogenous melatonin plays a crucial role in the growth and development of plants, enhancing plant antioxidant capacity, regulating osmotic substances, and oxidase activity. Research on various plant species like chrysanthemum, pear, and strawberry has demonstrated that exogenous melatonin boosts root activity, adjusts pH and permeable substances in the root system, and interacts with antioxidant enzyme activity, soluble protein content, and other endogenous substances to mitigate salt-alkali stress (*Tao et al., 2024*; *Wang, 2021*; *Ma, 2023*). Moreover, exogenous melatonin increases leaf antioxidant enzyme activity and chlorophyll content, leading to a reduction in leaf MDA content and effective alleviation of waterlogging stress. In a study by *Yang et al. (2023)*, eggplant seedlings showed significant changes in growth indicators after experiencing waterlogging stress compared to the control group. However, melatonin treatment notably restored the growth and development of eggplant seedlings to a certain extent. This restoration is

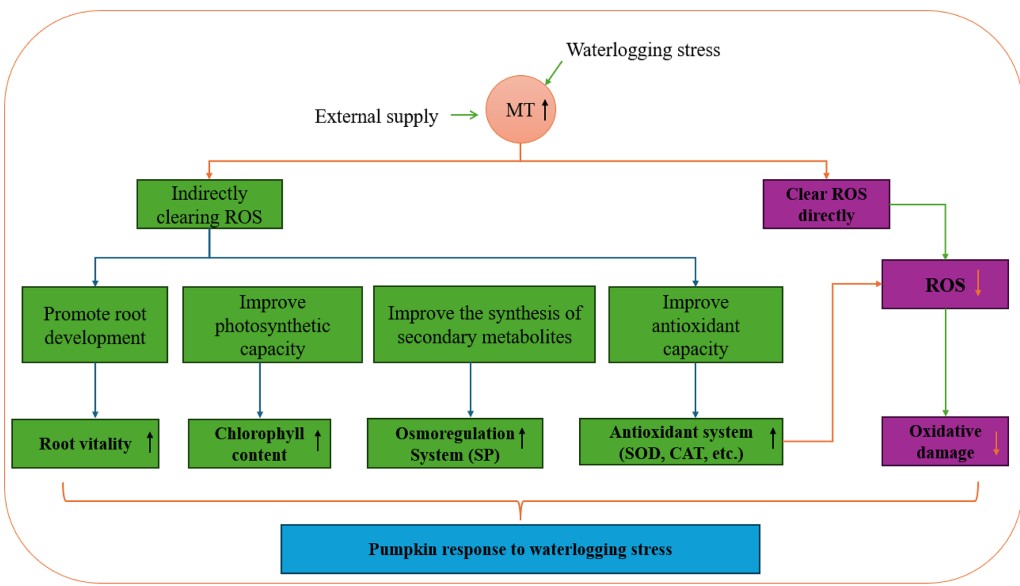

**Figure 12 Pumpkin response to waterlogging stress.**

attributed to melatonin's ability to enhance leaf photosynthesis by maintaining chlorophyll synthesis, increasing antioxidant enzyme activity, and organic osmotic adjustment substance content, thereby enhancing the scavenging capacity of reactive oxygen species and reducing membrane lipid peroxidation levels. Melatonin effectively minimizes the oxidative stress response triggered by adverse conditions and enhances plant physiological activities. Pre-stress application of exogenous melatonin has emerged as a cost-effective strategy to enhance plant stress resistance and mitigate yield losses due to stress (Fig. 12).

# CONCLUSION

Melatonin can effectively enhance the SOD and CAT activities of pumpkin seedlings under waterlogging stress, improving their ability to scavenge ROS and increase osmotic regulation substances. These effects alleviate the damage caused by waterlogging stress, maintain intracellular water levels and membrane system functions, and preserve cell turgor pressure. Additionally, melatonin enhances the photosynthetic capacity of pumpkin seedlings by increasing chlorophyll levels, enhancing root activity, and improving their overall tolerance to waterlogging stress. These beneficial effects are attributed to melatonin's ability to enhance antioxidant enzyme activity and increase the production of antioxidant substances such as ascorbic acid, glutathione, and carotenoids, ultimately reducing the ROS levels, to alleviate oxidative damage and enhance the resilience of pumpkin seedlings to waterlogging.

The growth of pumpkin seedlings was impeded under waterlogging stress, leading to a decrease in leaf chlorophyll content and root vigor, an increase in MDA content, and the accumulation of reactive oxygen species, which triggered higher activities of antioxidant enzymes. The application of exogenous melatonin increased chlorophyll content, enhanced the activities of antioxidant enzymes such as SOD and CAT, decreased lipid peroxidation, mitigated peroxidative damage, and stimulated pumpkin growth under

waterlogging stress. Specifically, treatment with 100 µmol·L$^{-1}$ melatonin exhibited superior efficacy in enhancing the waterlogging tolerance of pumpkins. The findings of this study can be applied to mitigate waterlogging damage in pumpkin cultivation, leading to increased economic benefits. Additionally, it lays a foundation for further research on the mechanisms of pumpkin tolerance to waterlogging.

## ACKNOWLEDGEMENTS

My deepest gratitude goes first and foremost to Professor Li, for constant encouragement and guidance. Without supporting of his project, this thesis could not have reached its present form.

### Funding
The research was supported by the Science and Technology Research Project of Henan Province (No. 242102110302). The funders had no role in study design, data collection and analysis, decision to publish, or preparation of the manuscript.

### Grant Disclosures
The following grant information was disclosed by the authors:
Science and Technology Research Project of Henan Province: 242102110302.

### Competing Interests
The authors declare that they have no competing interests.

### Author Contributions
- Zhenyu Liu conceived and designed the experiments, performed the experiments, analyzed the data, prepared figures and/or tables, authored or reviewed drafts of the article, and approved the final draft.
- Li Sun conceived and designed the experiments, analyzed the data, authored or reviewed drafts of the article, and approved the final draft.
- Zhenwei Liu conceived and designed the experiments, analyzed the data, prepared figures and/or tables, authored or reviewed drafts of the article, and approved the final draft.
- Xinzheng Li conceived and designed the experiments, authored or reviewed drafts of the article, and approved the final draft.

### Data Availability
The raw measurements are available in the Supplemental Files.

### Supplemental Information
Supplemental information for this article can be found online at http://dx.doi.org/10.7717/peerj.17927#supplemental-information.

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
