# Peer review of "Effect of exogenous melatonin on growth and antioxidant system of pumpkin seedlings under waterlogging stress"

_PeerJ, doi:10.7717/peerj.17927_

## Round 0.1 · original submission · Major Revisions

Please address concerns of all reviewers and amend manuscript accordingly.

·

Basic reporting

- Please rewrite the abstract with adding results details and add sentence to clear the importance of the findings

Experimental design

- Write the physiological indicators methods of determination with details

Validity of the findings

In results
- I suggest to add figure of correlations To clarify the extent of connection and relationships between treatments and results
- I suggest to add figure To clarify and conclude the experimental treatments with melatonin and parameters on pumpkin and the mechanism of action of melatonin and how it improved plant health under waterlogging

Conclusion
- I suggest adding a sentence to clarify the economic benefit from the experience and a future outlook for this work

Reviewer 2 ·

Basic reporting

The article has the potential for publication, however, it requires small changes (see attached file). The topic is interesting and very important. Exogenous application of melatonin has become one of the main alternatives to mitigate abiotic stress, such as flooding.

Experimental design

The methods used were described in detail. I suggest further detailing the physiological analysis carried out (enzymatic, chlorophyll content, and root activity).

Validity of the findings

No comment

Additional comments

I read with interest the manuscript entitled “Effect of exogenous melatonin on growth and antioxidant system of pumpkin seedlings under waterlogging stress”. In this study, was selected BM1, a flood-tolerant pumpkin variety, as the experimental material to explore the regulatory capacity of melatonin in alleviating the effect of waterlogging stress in pumpkins. The article's subject is important and relevant to the study area's scientific environment. Therefore, the manuscript needs some adjustments so that it can then be forwarded to the publication process. The manuscript has potential publication in this journal PeerJ and needs adjustments.

Annotated reviews are not available for download in order to protect the identity of reviewers who chose to remain anonymous.

Reviewer 3 ·

Basic reporting

Dear Editor,
Subject: Review for PeerJ (100164)
General comments
The present study, titled "Effect of exogenous melatonin on growth and antioxidant system of pumpkin seedlings under waterlogging stress" by Zhenyu Liu et al., investigated the impact of cover crops on soil microbial biomass and enzymatic activities, emphasizing the effects of melatonin in mitigating waterlogging stress in pumpkin seedlings is insightful and promising. It reveals that melatonin significantly improves growth and enhances the resilience of plants by boosting antioxidant enzyme activity and supporting osmotic regulation. The optimal concentration of melatonin, identified as 100 µmol·L-1, effectively enhances plant health by increasing chlorophyll content and overall photosynthetic capacity, proving its utility in enhancing agricultural resilience to environmental stresses. This research underscores the potential of melatonin as a valuable tool for improving crop tolerance to challenging conditions like waterlogging.
Overall, the introduction section need extensive revision, the materials and methods section is written with enough detailed.
I recommend a major revision and the manuscript can be accepted after incorporating these comments.

Experimental design

no comment

Validity of the findings

no comment

Additional comments

no comment

Annotated reviews are not available for download in order to protect the identity of reviewers who chose to remain anonymous.

---

## Round 0.2 · accepted · Accept

All issues pointed by the reviewers were addressed and the revised version is acceptable now.

·

Basic reporting

Authors made all of my suggestions and the manuscript is fit for publication

Experimental design

Authors made all of my suggestions and the manuscript is fit for publication

Validity of the findings

Authors made all of my suggestions and the manuscript is fit for publication

Reviewer 3 ·

Basic reporting

The authors have made the changes in the revised manuscript entitled “Effect of exogenous melatonin on growth and antioxidant system of pumpkin seedlings under waterlogging stress”. Now the revised version of the manuscript can be accepted for publication.

Experimental design

no

Validity of the findings

no

Additional comments

no